

# A Bayesian approach to construct confidence intervals for comparing the rainfall dispersion in Thailand

Patcharee Maneerat, Sa-aat Niwitpong and Suparat Niwitpong

Department of Applied Statistics, Faculty of Applied Science, King Mongkut's University of Technology North Bangkok, Bangkok, Thailand

## ABSTRACT

Natural disasters such as drought and flooding are the consequence of severe rainfall fluctuation, and rainfall amount data often contain both zero and positive observations, thus making them fit a delta-lognormal distribution. By way of comparison, rainfall dispersion may not be similar in enclosed regions if the topography and the drainage basin are different, so it can be evaluated by the ratio of variances. To estimate this, credible intervals using the highest posterior density based on the normal-gamma prior (HPD-NG) and the method of variance estimates recovery (MOVER) for the ratio of delta-lognormal variances are proposed. Monte Carlo simulation was used to assess the performance of the proposed methods in terms of coverage probability and relative average length. The results of the study reveal that HPD-NG performed very well and was able to meet the requirements in various situations, even with a large difference between the proportions of zeros. However, MOVER is the recommended method for equal small sample sizes. Natural rainfall datasets for the northern and northeastern regions of Thailand are used to illustrate the practical use of the proposed credible intervals.

## INTRODUCTION

Natural phenomena can often be random events in statistics, and natural rainfall is an important one because it is directly related to the quality of life, agriculture, economic growth, and industry, among others. Rice is the main export product from Thai agriculture, so natural rainfall is a crucial water resource for farmers. However, climate change has incurred prolonged droughts, thereby decreasing agricultural and fishery yields, and conversely, caused violent flooding and health-related issues in Thailand (*Marks, 2011*). Farmers who consume approximately 70% of the country's water supply are at the forefront of these impacts due to changes in rainfall amount (*Marks, 2011*). Thus, extreme rainfall variation is one of the main consequences of climate change and can result in drought or flooding, and ineffective water management can exacerbate both situations (*Duangdai & Likasiri, 2017*).

Corresponding author
Sa-aat Niwitpong,
sa-aat.n@sci.kmutnb.ac.th

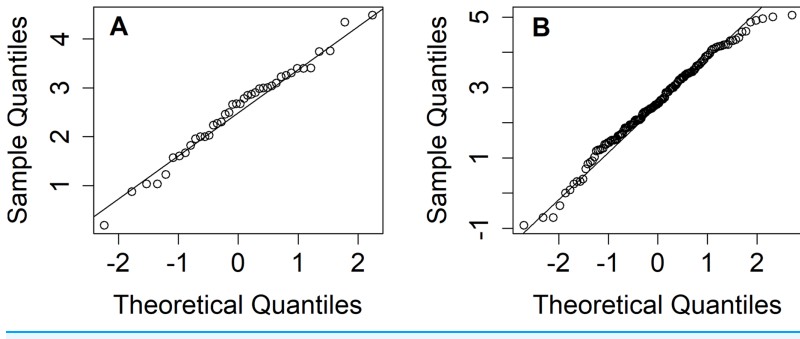

**Figure 1** Q–Q plots of log-transformation of non-zero records in (A) northern (B) northeastern areas.

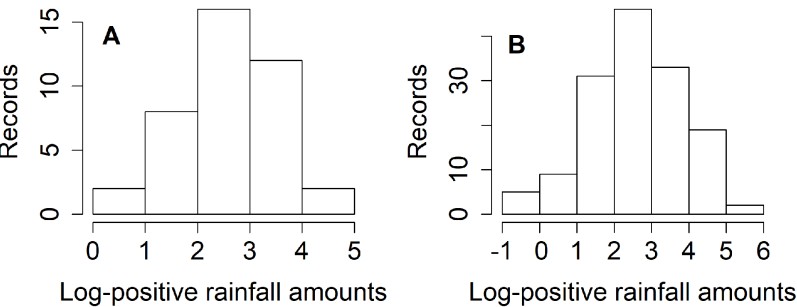

**Figure 2** Histogram plots of log-transformation of weekly positive rainfall records in (A) northern (B) northeastern areas.

Thailand is divided into five regions: northern, northeastern, central, eastern, and southern. The north is mountainous and contains the sources of several important rivers, including the Mekong Basin (a principal river system of Thailand). Nan is one of the provinces in the northern region faced with heavy rainfall for around two weeks which caused landslides and flooding during the rainy season (mid-May to mid-October) (*Hariraksapitak, Sriring & Navaratnam, 2018*). *Nicely & Counselor (2018)* reported that approximately 60% of the northeastern region's arable land has been turned over to total rice cultivation. Meanwhile, 5–10% of precipitation above the normal level occurred during May–July 2018 in the northern and northeastern regions, as reported by the Thai Meteorological Department (*Nicely & Counselor, 2018*). These situations have led us to consider estimating the rainfall fluctuation between the northern and northeastern regions of Thailand in terms of their ratio of variances during July 2018. It is of major importance to estimate the dispersion in weekly rainfall amount between the northern and northeastern regions in Thailand for the benefit of key industries, and the information gained could help in realizing climate change awareness. The information could be advantageous for the Thai government and other related organizations to realize and plan for solving or reducing the risk of environmental issues if a large variation in rainfall is known in advance. The results from normality plots (Fig. 1), histograms (Fig. 2), and applying the Akaike information criterion (AIC) show that the weekly natural rainfall data for both regions fit delta-lognormal distributions. Note that the log-transformed data of the positive rainfall amounts fit a normal distribution which is symmetrical.
*Aitchison & Brown (1963)* first introduced the delta-lognormal distribution for non-negative data containing zero values with the probability $0 < \delta < 1$; positive observations with the remainder of the probability $1 - \delta$ follow a lognormal distribution and the zeros follow a binomial distribution with binomial proportion $\delta$. The delta-lognormal distribution has been fitted for real-world examples in many research areas such as the environment (*Owen & DeRouen, 1980*; *Hasan & Krishnamoorthy, 2018*), fishery surveys (*Pennington, 1983*; *Smith, 1988*, *1990*; *Lo, Jacobson & Squire, 1992*; *Fletcher, 2008*; *Wu & Hsieh, 2014*) and medicine (*Zhou & Tu, 1999*, *2000*; *Tian & Wu, 2006*; *Hasan & Krishnamoorthy, 2018*).

In statistical inference, one of the parameters of interest is the variance (dispersion in environment) defined as the second central moment; the positive square root of the variance is called the standard deviation (*Casella & Berger, 2002*). To compare two independent populations, the ratio of their variances is the measurement of the variation between them with no difference resulting in a ratio equal to 1. Confidence intervals (CIs) have been applied to several distributions to estimate the variability in terms of variance and the ratio of variances. For example, *Krishnamoorthy, Mathew & Ramachandran (2006)* obtained CIs for lognormal variance based on the idea of the generalized confidence interval (GCI) which they applied to the variation in the air lead levels in 15 industrial facilities in a health hazard evaluation. *Bebu & Mathew (2008)* claimed that GCI was better than a modified signed log-likelihood ratio test for establishing CIs for the ratio of variances for bivariate lognormal distributions even when the sample sizes were small. *Niwitpong (2017)* showed that GCI performed well for the ratio of variances of lognormal distributions to solve the problems that occur when applying the traditional approach.

*Casella & Berger (2002)* argued that CIs can capture the parameter of interest better than point estimates. Although a few attempts have been conducted on CIs for the ratio of variances for some distributions, research on the delta-lognormal distribution has yet to be carried out. Clearly, there is a need for research that investigates and constructs CIs for the ratio of delta-lognormal variances using the highest posterior density based on normal-gamma prior (HPD-NG) and MOVER. These proposed CIs were compared with the existing HPD-based Jeffreys' (HPD-Jef) and Jeffreys' Rule (HPD-Rul) priors of *Harvey & Van der Merwe (2012)*, GCI of *Wu & Hsieh (2014)* and fiducial GCI (FGCI) of *Hasan & Krishnamoorthy (2018)*.

The organization of this paper is as follows. The concepts of all of the CIs for the ratio of variances in delta-lognormal distributions are elaborated in "Methods". The simulation procedure and numerical results are reported in "Results." In "Discussion," the proposed methods are applied to real-world datasets to estimate the rainfall fluctuation between the northern and northeastern regions in Thailand. Last, we present a discussion and conclusions on the study outcomes.

## METHODS

Let $X_{ij} = (X_{i1}, X_{i2}, \ldots, X_{ini})$; $i = 1, 2$ and $j = 1, 2, \ldots, n_i$ be non-negative random samples draw from a delta-lognormal distribution with parameters $\mu_i$, $\sigma_i^2$ and $\delta_i$, stand for

---

**Algorithm 1.**

(1) Generate $W_i \sim N(0, 1)$, $Z_i \sim N(0, 1)$ and $U_i \sim \chi^2_{n_{i(1)}-1}$.

(2) Compute $R_{\mu_i}$, $R_{\sigma_i^2}$, and $R_{\delta_i}$.

(3) Compute $R_{\ln \omega_i}$.

(4) Compute $R_{\theta_i}$.

(5) Repeat 1–4 a number of times (say, $m = 2,500$).

(6) For the 2,500 times, compute $(100 - \zeta)\%$GCI for $\theta$.

---

$X_{ij} \sim \Delta(\mu_i, \sigma_i^2, \delta_i)$. For $X_{ij} = 0$, the number of having zero $n_{i(0)} \sim B(n_i, \delta_i)$ and $n_i = n_{i(0)} + n_{i(1)}$. For $X_{ij} > 0$, $X_{ij} \sim LN(\mu_i, \sigma_i^2)$ where $Y_{ij} = \ln(X_{ij}) \sim N(\mu_i, \sigma_i^2)$. *Aitchison & Brown (1963)* defined the population variance of $X_{ij}$ as

$$\omega_i = (1 - \delta_i) \exp(2\mu_i + \sigma_i^2) \left[\exp(\sigma_i^2) - (1 - \delta_i)\right] \tag{1}$$

The maximum likelihood estimates of $\delta_i$, $\mu_i$ and $\sigma_i^2$ are $\hat{\delta}_i = \frac{n_{i(0)}}{n_i}$, $\hat{\mu}_i = \frac{1}{n_{i(1)}}\sum_{j=1}^{n_{i(1)}} y_{ij}$, and $\hat{\sigma}_i^2 = \frac{1}{n_{i(1)}}\sum_{j=1}^{n_{i(1)}} \left(y_{ij} - \hat{\mu}_i\right)^2$, respectively. From Eq. (1), the ratio in delta-lognormal variances is obtained as

$$\theta = \ln \omega_1 - \ln \omega_2 \tag{2}$$

where $\omega_i$ is log-transformed as $\ln \omega_i = \ln(1 - \delta_i) + (2\mu_i + \sigma_i^2) + \ln\left[\exp(\sigma_i^2) - (1 - \delta_i)\right]$. The methods for constructing CIs for $\theta$ are described as seen below.

## GCI

On the ideas of GCI, the generalized pivotal quantity (GPQ) is necessary to satisfy the two requirements of *Weerahandi (1993)*. The GPQ of $\delta_i$ based on VST was proposed by *Wu & Hsieh (2014)* as

$$R_{\delta_i} = \sin^2\left[\arcsin \sqrt{\hat{\delta}_i} - \frac{W_i}{2\sqrt{n_i}}\right] \tag{3}$$

where $W_i = 2\sqrt{n_i}\left(\arcsin\sqrt{\hat{\delta}_i} - \arcsin\sqrt{\delta_i}\right) \overset{d}{\sim} N(0, 1)$. The GPQs of $\mu_i$ and $\sigma_i^2$ are

$$\begin{aligned} R_{\mu_i} &= \hat{\mu}_i - Z_i\sqrt{(n_{i(1)} - 1)\hat{\sigma}^2/[n_{i(1)}U_i]} \\ R_{\sigma_i^2} &= (n_{i(1)} - 1)\hat{\sigma}_i^2/U_i \end{aligned} \tag{4}$$

which are presented by *Krishnamoorthy & Mathew (2003)*. These GPQs lead to obtain the GPQ of $\theta$ as

$$R_\theta = R_{\ln \omega_1} - R_{\ln \omega_2} \tag{5}$$

where $R_{\ln \omega_i} = \ln(1 - R_{\delta_i}) + (2R_{\mu_i} + R_{\sigma_i^2}) + \ln\left[\exp(R_{\sigma_i^2}) - (1 - R_{\delta_i})\right]$.
$[R_\theta(\zeta/2), R_\theta(1 - \zeta/2)]$ becomes the $100(1 - \zeta)\%$GCI for $\theta$. Algorithm 1 shows the steps to compute GCI as seen above.

---

> **Algorithm 2.**
>
> (1) Generate $Z_i \sim N(0, 1)$ and $U_i \sim \chi^2_{n_{i(1)}-1}$.
>
> (2) Compute $T_{\mu_i}$, $T_{\sigma_i^2}$, and $T_{1-\delta i}$.
>
> (3) Compute $T_{\ln \omega_i}$.
>
> (4) Compute $T_{\theta_i}$.
>
> (5) Repeat 1–4 a number of times (say, $m = 2{,}500$).
>
> (6) For the 2,500 times, compute $(100 - \zeta)\%$FGCI for $\theta$.

## FGCI

The fiducial generalized pivotal quantity (FGPQ) was proved and defined by *Hannig, Iyer & Patterson (2006)*. *Hannig (2009)* claimed that their generalized fiducial recipe has been developed as the GPQ concept, so their ideas are applied with generalized inference directly. Here $X_{ij} \sim \Delta(\mu_i, \sigma_i^2, \delta_i)$ is considered. *Hasan & Krishnamoorthy (2018)* showed the FGPQs of $\mu_i$ and $\sigma_i^2$ as

$$T_{\mu_i} = \hat{\mu}_i - Z_i \sqrt{\hat{\sigma}_i^2/(n_{i(1)} U_i)} \tag{6}$$

$$T_{\sigma_i^2} = \hat{\sigma}_i^2/U_i \tag{7}$$

where $Z_i \sim N(0, 1)$ and $U_i \sim \chi^2_{ni(1)-1}/(n_{i(1)} - 1)$ are independent random variables. Also, the FGPQs of $\delta_i$ was developed as $T_{1-\delta i} \sim beta(n_{i(1)} + 0.5, n_{i(0)} + 0.5)$. By three FGPQs, the FGPQ of $\ln \omega_i$ is defined as $T_{\ln \omega_i} = \ln T_{1-\delta_i} + (2T_{\mu_i} + T_{\sigma_i^2}) + \ln\left[\exp(T_{\sigma_i^2}) - T_{1-\delta_i}\right]$, then the FGPQs of $\theta$ can be expressed as

$$T_\theta = T_{\ln \omega_1} - T_{\ln \omega_2} \tag{8}$$

which can establish the $100(1-\zeta)\%$FGCI for $\theta$ that is $[T_\theta(\zeta/2), T_\theta(1 - \zeta/2)]$; $T_\theta(\zeta)$ stands for $(\zeta)$100th percentile of $T_\theta$. Algorithm 2 details the computation of FGCI.

## MOVER interval

This interval is expanded from CIs for delta-lognormal mean of *Hasan & Krishnamoorthy (2018)* to the ratio of delta-lognormal variances. Given the individual CIs for the parameters of interest $\gamma$, the MOVER concept is used to establish a CI for a linear combination of parameters, proposed by *Krishnamoorthy & Oral (2017)*.

Let $\gamma = (\gamma_1, \gamma_2, \ldots \gamma_p)$ be a p-dimensional parameter vector. The estimate of $\gamma_i$ is $\hat{\gamma}_i; i = 1, 2, \ldots, p$ that are independent. The MOVER interval for $\sum_{i=1}^p c_i \gamma_i$ is defined as

$$CI_{\text{MOVER}} = \left[ \sum_{i=1}^p c_i \hat{\gamma}_i - \sqrt{\sum_{i=1}^p c_i^2 (\hat{\gamma}_i - l_i^*)^2}, \sum_{i=1}^p c_i \hat{\gamma}_i + \sqrt{\sum_{i=1}^p c_i^2 (\hat{\gamma}_i - u_i^*)^2} \right] \tag{9}$$

where $l_i^* = l_i$ if $c_i > 0$, $u_i$ if $c_i < 0$ and $u_i^* = u_i$ if $c_i > 0$, $l_i$ if $c_i < 0$. Moreover, $(l_i, u_i)$ stands for the CI for $\gamma_i$. Here we let

$$\ln \omega_i = \omega_i^{(1)} + \omega_i^{(2)} + \omega_i^{(3)} = \ln(1-\delta_i) + (2\mu_i + \sigma_i^2) + \ln\left[\exp(\sigma^2) - (1-\delta_i)\right] \tag{10}$$

Using $\hat{\mu}_i, \hat{\sigma}_i^2, \hat{\delta}_i$ from a sample, $\ln \hat{\omega}_i = \hat{\omega}_i^{(1)} + \hat{\omega}_i^{(2)} + \hat{\omega}_i^{(3)}$ is obtained, and CIs for individual parameters $\omega_i^{(1)}$, $\omega_i^{(2)}$ and $\omega_i^{(3)}$ are also constructed. To begin with $\omega_i^{(1)}$, $(100 - \zeta)\%$ CI-based Wilson was proposed by *Wilson (1927)*, that is

$$CI_{\omega_i^{(1)}} = [l_{\omega_i^{(1)}}, u_{\omega_i^{(1)}}] = \ln\left[\left\{(n_{i(1)} + Z_{i,\frac{\zeta}{2}}^2/2) \mp \left(Z_{i,1-\frac{\zeta}{2}}\sqrt{\frac{n_{i(0)}n_{i(1)}}{n_i} + \frac{Z_{i,\frac{\zeta}{2}}^2}{4}}\right)\right\}/(n_i + Z_{i,\frac{\zeta}{2}}^2)\right] \tag{11}$$

where $Z_i = \frac{n_{i(1)} - n_i(1-\delta_i)}{\sqrt{n_i\delta_i(1-\delta_i)}} \overset{d}{\sim} N(0,1)$. The $(100 - \zeta)\%$CI for $\omega_i^{(2)}$ is developed from *Zou, Taleban & Huo (2009)*, then

$$
\begin{aligned}
CI_{\omega_i^{(2)}} &= [l_{\omega_i^{(2)}}, u_{\omega_2^{(1)}}] \\
&= \left[\hat{\omega}_i^{(2)} - \sqrt{\frac{4\hat{\sigma}_i^2 T_i^2}{n_{i(1)}} + \hat{\sigma}_i^4\left(1 - \frac{n_{i(1)} - 1}{\chi_{1-\zeta/2,n_{i(1)}-1}^2}\right)^2}, \hat{\omega}_i^{(2)} + \sqrt{\frac{4\hat{\sigma}^{2i} T_i^2}{n_{i(1)}} + \hat{\sigma}_i^4\left(\frac{n_{i(1)} - 1}{\chi_{\zeta/2,n_{i(1)}-1}^2} - 1\right)^2}\right]
\end{aligned} \tag{12}
$$

where $T_i = \frac{\hat{\mu}_i - \mu_i}{\sqrt{\sigma_i^2/n_{i(1)}}} \overset{d}{\sim} N(0,1)$ and $\chi_{ni(1)-1}^2$ denoted as chi-square distribution with $n_{i(1)} - 1$ degree of freedom. Next, the $(100 - \zeta)\%$CI for $\omega_i^{(3)}$ is proposed as

$$CI_{\omega_i^{(3)}} = [l_{\omega_i^{(3)}}, u_{\omega_i^{(3)}}] \tag{13}$$

where

$$l_{\omega_i^{(3)}} = \ln\left[\hat{\omega}_i^{(3)} - \sqrt{\exp\left\{\hat{\sigma}_i^4\left(1 - \frac{n_{i(1)} - 1}{\chi_{1-\zeta/2,n_{i(1)}-1}^2}\right)^2\right\} + \left\{\frac{Z_{i,1-\frac{\zeta}{2}}\sqrt{\frac{n_{i(0)}n_{i(1)}}{n_i} + \frac{Z_{i,\frac{\zeta}{2}}^2}{4}}}{n_i + Z_{i,\frac{\zeta}{2}}^2}\right\}^2}\right]$$

$$u_{\omega_i^{(3)}} = \ln\left[\hat{\omega}_i^{(3)} + \sqrt{\exp\left\{\hat{\sigma}_i^4\left(\frac{n_{i(1)} - 1}{\chi_{\zeta/2,n_{i(1)}-1}^2} - 1\right)^2\right\} + \left\{\frac{Z_{i,1-\frac{\zeta}{2}}\sqrt{\frac{n_{i(0)}n_{i(1)}}{n_i} + \frac{Z_{i,\frac{\zeta}{2}}^2}{4}}}{n_i + Z_{i,\frac{\zeta}{2}}^2}\right\}^2}\right]$$

As mentioned above, the $(100 - \zeta)\%$ MOVER interval for $\ln \omega_i$ can be written as

$$CI_{\ln \omega_i} = [L_{\ln \omega_i}, U_{\ln \omega_i}] \tag{14}$$

where

$$L_{\ln \omega_i} = \left(\hat{\omega}_i^{(1)} + \hat{\omega}_i^{(2)} + \hat{\omega}_i^{(3)}\right) - \sqrt{\left(\hat{\omega}_i^{(1)} - l_{\omega_i^{(1)}}\right)^2 + \left(\hat{\omega}_i^{(2)} - l_{\omega_i^{(2)}}\right)^2 + \left(\hat{\omega}_i^{(3)} - l_{\omega_i^{(3)}}\right)^2}$$

> **Algorithm 3.**
>
> (1) Generate $Z_i \sim N(0, 1)$, $T_i \sim N(0, 1)$ and $U_i \sim \chi^2_{n_{i(1)}-1}$ are independent.
>
> (2) Compute $CI_{\omega_i^{(1)}}$, $CI_{\omega_i^{(2)}}$ and $CI_{\omega_i^{(3)}}$.
>
> (3) Compute $CI_{\ln \omega i}$ and $CI_\theta$.
>
> (4) Compute $(100 - \zeta)\%$MOVER for $\theta$.

$$U_{\ln \omega_i} = \left( \hat{\omega}_i^{(1)} + \hat{\omega}_i^{(2)} + \hat{\omega}_i^{(3)} \right) + \sqrt{\left( \hat{\omega}_i^{(1)} - u_{\omega_i^{(1)}} \right)^2 + \left( \hat{\omega}_i^{(2)} - u_{\omega_i^{(2)}} \right)^2 + \left( \hat{\omega}_i^{(3)} - u_{\omega_i^{(3)}} \right)^2}$$

Therefore, the $(100 - \zeta)\%$ MOVER interval for $\theta$ is based on *Donner & Zou (2012)*, given by

$$CI_\theta = [L_\theta, U_\theta] \tag{15}$$

where

$$L_\theta = (\ln \hat{\omega}_1 - \ln \hat{\omega}_2) - \sqrt{(\ln \hat{\omega}_1 - L_{\ln \omega_1})^2 + (U_{\ln \omega_2} - \ln \hat{\omega}_2)^2}$$

$$U_\theta = (\ln \hat{\omega}_1 - \ln \hat{\omega}_2) + \sqrt{(U_{\ln \omega_1} - \ln \hat{\omega}_1)^2 + (\ln \hat{\omega}_2 - L_{\ln \omega_2})^2}$$

The following detail is used to compute the MOVER interval.

## HPD credible interval

The HPD credible interval is a parameter estimate based on the posterior probability in Bayesian framework. *Box & Tiao (1973)* defined the HPD ideas consisting of two requirements: the set of value that contain $100(1 - \zeta)\%$ of the posterior distribution and the property that the density within the region is equal or greater than outside. The posterior density is unimodal and symmetric, the region based on their definition becomes to a equal-sided CI ($\zeta/2$ and $1 - \zeta/2$ percentiles of the posterior density). This is clearly different between equal-sided CI and HPD credible interval if there is a highly skewed in the posterior density. Recently, HPDs based on beta and uniform priors were recommended to constructed for single mean and the difference between two delta-lognormal means proposed by *Maneerat, Niwitpong & Niwitpong (2019)*. The HPD credible interval is then focused on our study. Recall that $X_{ij} > 0$ be random variables from lognormal distribution with parameter $\kappa = \left( \mu_1 \quad \sigma_1^2 \quad \mu_2 \quad \sigma_2^2 \right)'$; $Y_{ij} = \ln X_{ij} \sim N(\mu_i, \sigma_i^2)$. The Fisher information matrix of $\kappa$ is $I(\kappa) = diag\left[ \frac{n_{1(1)}}{\sigma_1^2} \quad \frac{n_{1(1)}}{2\sigma_1^4} \quad \frac{n_{2(1)}}{\sigma_2^2} \quad \frac{n_{2(1)}}{2\sigma_2^4} \right]$. For $X_{ij} = 0$, the number of zero values $n_{i(0)}$ be random sample of binomial distribution, denoted as $B(n_i, \delta_i)$. The likelihood function of $X_{ij}$ is

$$P(x|\omega) \propto \prod_{i=1}^{2} \delta_i^{n_{i(0)}} (1 - \delta_i)^{n_{i(1)}} (\sigma_i^2)^{-n_{i(1)}/2} \exp\left\{ -\frac{1}{2\sigma_i^2} \sum_{j=1}^{n_{i(1)}} (\ln x_i - \mu_i)^2 \right\} \tag{16}$$

which leads to obtain the Fisher information matrix of $\omega = \begin{pmatrix} \mu_1 & \sigma_1^2 & \delta_1 & \mu_2 & \sigma_2^2 & \delta_2 \end{pmatrix}'$

$$I(\omega) = diag\left[ \frac{n_{1(1)}}{\sigma_1^2} \quad \frac{n_{1(1)}}{2\sigma_1^4} \quad \frac{n_1}{\delta_1(1-\delta_1)} \quad \frac{n_{2(1)}}{\sigma_2^2} \quad \frac{n_{2(1)}}{2\sigma_2^4} \quad \frac{n_2}{\delta_2(1-\delta_2)} \right]$$

Here $\theta = \ln \omega_1 - \ln \omega_2$ so that the HPD credible intervals based on different priors for $\theta$ are described.

### Jeffreys' prior

The Jeffreys' prior for $\omega$ is defined as

$$P(\omega)_J \propto \prod_{i=1}^{2} P(\kappa)_J P(\delta) \propto \prod_{i=1}^{2} \sigma_i^{-2}[\delta_i(1-\delta_i)]^{-1/2} \tag{17}$$

where $P(\kappa)_J \propto \sqrt{I(\sigma^2)}$ and $P(\delta)_J \propto \sqrt{I(\delta)}$. The prior Eq. (17) is combined with its likelihood Eq. (16) to obtain the posterior distributions of each parameters $\sigma_i^2$, $\mu_i$, and $\delta_i$. Firstly, the posterior density of $\omega$ is using Bayes' theorem (*Casella & Berger, 2002*), given by

$$P(\omega|x)_J \propto \prod_{i=1}^{2} \delta_i^{n_{i(0)}-1/2}(1-\delta_i)^{n_{i(1)}-1/2}\sigma_i^{-(n_{i(1)}+2)} \exp\left\{ -\frac{1}{2\sigma_i^2}\left[ (n_{i(1)}-1)\hat{\sigma}_i^2 + n_{i(1)}(\hat{\mu}_i - \mu_i)^2 \right] \right\} \tag{18}$$

Obtain that

$$P(\sigma_i^2|x)_J \propto \prod_{i=1}^{2} (\sigma_i^2)^{-\left( \frac{n_{i(1)}-1}{2}+1 \right)} \exp\left\{ -\frac{(n_{i(1)}-1)\hat{\sigma}_i^2}{2\sigma_i^2} \right\} \tag{19}$$

which is the posterior density of $\sigma^2|x$, as inverted gamma distribution, denoted as $IG(a_i, b_i)$; $a_i = r_i/2$ and $b_i = r_i\hat{\sigma}_i^2/2$; $r_i = n_{i(1)} - 1$. Given $\sigma_i^2$ and $x$, one obtains that

$$P(\mu_i|\sigma_i^2, x)_J \propto \prod_{i=1}^{2} \exp\left\{ -\frac{n_{i(1)}}{2\sigma_i^2}(\mu_i - \hat{\mu}_i)^2 \right\} \tag{20}$$

This is the posterior density of $\mu_i|\sigma_i^2, x$, as normal $N(\hat{\mu}_i, \sigma_i^2/n_{i(1)})$. For $\delta_i$, its posterior was proved as

$$p(\delta_i|x)_J \propto \delta_i^{(n_{i(0)}+1/2)-1}(1-\delta i)^{(n_{i(1)}+1/2)-1}$$

which is the beta distribution with $c_i$ and $d_i$, denoted as $beta(c_i, d_i)$; $c_i = n_{i(0)} + 1/2$ and $d_i = n_{i(1)} + 1/2$.

### Jeffreys' Rule prior

According to *Harvey & Van der Merwe (2012)*, the difference between Jeffreys and Jeffreys' Rule priors is the posterior densities of $\sigma_i^2$ and $\delta_i$, while Jeffreys' Rule prior is defined as

$$P(\omega)_{JR} \propto \prod_{i=1}^{2} \sigma_i^{-3}\delta_i^{-1/2}(1-\delta_i)^{1/2} \tag{21}$$

**Algorithm 4.**

(1) Generate $\sigma_{i}^{2*}$ denoted as the posterior distribution of $\sigma_{i}^{2}$ based on priors:
  - Jeffreys' prior: $\sigma_{i(J)}^{2*} \sim IG(r_i/2, r_i\hat{\sigma}_i^2/2)$; $r_i = n_{i(1)} - 1$.
  - Jeffreys' Rule prior: $\sigma_{i(JR)}^{2*} \sim IG(s_i/2, s_i\hat{\sigma}_i^2/2)$; $s_i = n_{i(1)} + 1$.
  - NG prior: $\sigma_{i(NG)}^{2*} \sim IG(\sigma_i^2|\alpha_{i,ni(1)}, \beta_{i,ni(1)})$.

(2) Given $\sigma_{i}^{2*}$ and $x$, generate $\mu_i^*$ depends on the following priors:
  - Jeffreys' prior: $\mu_{i(J)}^* \sim N(\hat{\mu}_i, \sigma_{i(J)}^{2*}/n_{i(1)})$.
  - Jeffreys' Rule prior: $\mu_{i(JR)}^* \sim N(\hat{\mu}_i, \sigma_{i(JR)}^{2*}/n_{i(1)})$.
  - NG prior: $\mu_{i(NG)}^* \sim t_{df}(\mu_i|\mu_{i,n_{i(1)}}, \beta_{i,n_{i(1)}}/[\alpha_{i,n_{i(1)}}k_{i,n_{i(1)}}])$.

(3) Generate $\delta_i^*$,
  - Jeffreys' prior: $\delta_{i(J)}^* \sim beta(n_{i(0)} + 1/2, n_{i(1)} + 1/2)$.
  - Jeffreys' Rule prior: $\delta_{i(JR)}^* \sim beta(n_{i(0)} + 1/2, n_{i(1)} + 3/2)$.
  - NG prior: $\delta_{i(NG),ni(1)}^* \sim beta(n_{i(1)} + d_i, n_{i(0)} + d_i)$.

(4) Compute $\omega_i^*$,
  - Jeffreys' prior: $\omega_{i(J)}^* = (1 - \delta_{i(J)}^*) \exp\left(2\mu_{i(J)}^* + \sigma_{i(J)}^{2*}\right)\left[\exp(\sigma_{i(J)}^{2*}) - (1 - \delta_{i(J)}^*)\right]$.
  - Jeffreys' Rule prior: $\omega_{i(JR)}^* = (1 - \delta_{i(JR)}^*) \exp\left(2\mu_{i(JR)}^* + \sigma_{i(JR)}^{2*}\right)\left[\exp(\sigma_{i(JR)}^{2*}) - (1 - \delta_{i(JR)}^*)\right]$.
  - NG prior: $\omega_{i(NG)}^* = \delta_{i(NG),n_{i(1)}}^* \exp\left(2\mu_{i(NG)}^* + \sigma_{i(NG)}^{2*}\right)\left[\exp(\sigma_{i(NG)}^{2*}) - \delta_{i(NG),n_{i(1)}}^*\right]$.

(5) Compute $\theta^* = \ln \omega_1^* - \ln \omega_2^*$ based on three priors.

(6) Repeat 1–5 a number of times (say, $m = 2{,}500$).

(7) For the 2500 times, compute $(100 - \zeta)\%$HPD interval for $\theta$ in each prior.

To obtain the joint posterior of $\omega$ denoted as $P(\omega|x)_{JR}$, the Jeffreys' Rule prior Eq. (21) is combined with the likelihood Eq. (16). For $\sigma_i^2$, its posterior has been changed to inverted gamma distribution with shape parameter $s_i/2$ and scale parameter $s_i\hat{\sigma}_i^2/2$; $s_i = n_{i(1)} + 1$. Then, the posterior distribution of $\mu_i$ given $\sigma_i^2$ and $x$ is changed because it depends on the posterior density of $\sigma_i^2$. Likewise, the posterior of $\delta_i$ becomes $P(\delta|x)_{JR} = beta(n_{i(0)} + 1/2, n_{i(1)} + 3/2)$. Their posterior distributions represent to estimate own parameter, meanwhile $P(\sigma_i^2|x_{ij})_{JR}$, $P(\mu_i|\sigma_i^2, x_{ij})_{JR}$ and $P(\delta_i|x_{ij})_{JR}$ are independent. As a result, HPD credible interval for $\theta$ is computed based on Jeffreys' Rule prior. The steps for establishing HPD-intervals based on two priors are detailed as Algorithm 4.

### Normal-gamma prior

*DeGroot (1970)* defined the conjugate families for random sample of normal distribution. This is a necessary to develop credible interval based on Bayesian approach for the delta-lognormal variance. Using Theorem 1 of the conjugate families for random sample of normal distribution (*DeGroot, 1970*), assume that ($\mathbf{Y} = \ln X_{ij}$; $i = 1, 2$ $j = 1, 2, \ldots, n_{i(1)}$) be a random sample from normal distribution with the mean $\mu = (\mu_1, \mu_2)$ and precision $\lambda = (\lambda_1, \lambda_2)$; $\lambda_i = 1/\sigma_i^2$ where $X \sim LN(\mu, \lambda)$. The normal-gamma prior of $\tau = (\mu, \lambda)'$ is defined the marginal distributions between normal $\mu_i|\lambda_i \sim N(\mu, [k_i\lambda_i]^{-1})$ and gamma $\lambda_i \sim G(\alpha_i, \beta_i)$, given by

$$P(\tau) = P(\lambda; \alpha_{(0)}, \beta_{(0)})P(\mu|\lambda; \mu_{(0)}, k_{(0)}\lambda)$$

$$= \prod_{i=1}^{2}\left[\frac{\beta_{i(0)}}{\Gamma(\alpha_{i(0)})}\lambda_i^{\alpha_{i(0)}-1}\exp\left\{-\beta_{i(0)}\lambda_i\right\}\right]\left[\frac{(k_{i(0)}\lambda_i)^{1/2}}{\sqrt{2\pi}}\exp\left\{-\frac{k_i\lambda_i}{2}(\mu_i-\mu_{i(0)})^2\right\}\right] \quad (22)$$

$$\propto \prod_{i=1}^{2}\lambda_i^{\alpha_{i(0)}-1/2}\exp\left\{\frac{-\lambda_i}{2}\left[2\beta_{i(0)}+k_{i(0)}(\mu_i-\mu_{i(0)})^2\right]\right\}$$

The likelihood can be written as

$$P(y;\tau) = \prod_{i=1}^{2}(2\pi)^{-n_{i(1)}/2}\lambda_i^{n_{i(1)}/2}\exp\left\{\frac{-\lambda_i}{2}\sum_{i=1}^{n_{i(1)}}(y_{ij}-\mu_i)^2\right\}$$

$$\propto \prod_{i=1}^{2}\lambda_i^{n_{i(1)}/2}\exp\left\{\frac{-\lambda_i}{2}\left[\sum_{i=1}^{n_{i(1)}}(y_{ij}-\hat{\mu}_i)^2+n_{i(1)}(\hat{\mu}-\mu_i)^2\right]\right\} \quad (23)$$

The posterior distribution of $\tau$ can be derived as

$$P(\tau|y) = P(y;\tau)P(\tau)$$

$$\propto \prod_{i=1}^{2}\lambda_i^{\frac{n_{i(1)}}{2}}\exp\left\{\frac{-\lambda_i}{2}\left[\sum_{i=1}^{n_{i(1)}}(y_{ij}-\hat{\mu}_i)^2+n_{i(1)}(\hat{\mu}-\mu_i)^2\right]\right\}\lambda_i^{\alpha_{i(0)}-\frac{1}{2}}\exp\left\{-\lambda\beta_{i(0)}\right\}$$

$$\exp\left\{\frac{-\lambda_i}{2}\left[k_{i(0)}(\mu_i-\mu_{i(0)})^2\right]\right\}$$

$$\propto \prod_{i=1}^{2}\lambda^{\frac{n_{i(1)}}{2}+\alpha_{i(0)}-\frac{1}{2}}\exp\left\{-\lambda_i\left[\frac{1}{2}\sum_{i=1}^{n_{i(1)}}(y_{ij}-\hat{\mu}_i)^2+\beta_{i(0)}\right]\right\}$$

$$\exp\left\{\frac{-\lambda_i}{2}\left[n_{i(1)}(\hat{\mu}-\mu_i)^2+k_{i(0)}(\mu_i-\mu_{i(0)})^2\right]\right\}$$

$$\propto \prod_{i=1}^{2}\lambda_i^{\alpha_i^*-\frac{1}{2}}\exp\left\{-\lambda_i\left[\frac{1}{2}\sum_{i=1}^{n_{i(1)}}(y_{ij}-\hat{\mu}_i)^2+\beta_{i(0)}\right]\right\} \quad (24)$$

$$\exp\left\{\frac{-\lambda_i}{2}\left[(n_{i(1)}+k_{i(0)})(\mu_i-\mu_i^*)^2+\frac{n_{i(1)}k_{i(0)}[\hat{\mu}_i-\mu_{i(0)}]^2}{n_{i(1)}+k_{i(0)}}\right]\right\}$$

$$\propto \prod_{i=1}^{2}\lambda_i^{\alpha_i^*-1}\exp\left\{-\lambda_i\left[\beta_{i(0)}+\frac{1}{2}\sum_{i=1}^{n_{i(1)}}(y_{ij}-\hat{\mu}_i)^2+\frac{n_{i(1)}k_{i(0)}[\hat{\mu}_i-\mu_{i(0)}]^2}{2(n_{i(1)}+k_{i(0)})}\right]\right\}$$

$$\lambda_i^{\frac{1}{2}}\exp\left\{\frac{-\lambda_i[n_{i(1)}+k_{i(0)}]}{2}(\mu_i-\mu_i^*)^2\right\}$$

$$\propto \prod_{i=1}^{2}\lambda_i^{\alpha_i^*-1}\exp\left\{\lambda_i\beta_i^*\right\}\lambda_i^{\frac{1}{2}}\exp\left\{\frac{-\lambda_i k_i^*}{2}(\mu_i-\mu_i^*)^2\right\}$$

This implied that $\tau_i|y \sim Normal(\mu_i|\mu_i^*, [k_i^*\lambda]^{-1})Gamma(\lambda_i|\alpha_i^*, \beta_i^*)$ where

$\alpha_i^* = \alpha_{i(0)}+n_{i(1)}/2$, $\beta_i^* = \beta_{i(0)}+\frac{1}{2}\sum_{i=1}^{n_{i(1)}}(y_{ij}-\hat{\mu}_i)^2+\frac{n_{i(1)}k_{i(0)}[\hat{\mu}_i-\mu_{i(0)}]^2}{2(n_{i(1)}+k_{i(0)})}$, $k_i^* = n_{i(1)}+k_{i(0)}$,

and $\mu_i^* = \frac{\mu_{i(0)}k_{i(0)}+n_{i(1)}\hat{\mu}_i}{n_{i(1)}+k_{i(0)}}$. From Eq. (22), the normal-gamma prior of $\tau$ is defined as

$$P(\tau) \propto \prod_{i=1}^{2}\lambda_i^{-1} \quad (25)$$

which is normal-gamma distribution, denoted as $NG(\mu_i, \lambda_i | \mu, k_{i(0)} = 0, \alpha_{i(0)} = -1/2, \beta_{i(0)} = 0$ so that its posterior of $\tau$ is derived from Eq. (24) that becomes

$$P(\tau|y) \propto \prod_{i=1}^{2} \lambda_i^{\frac{n_{i(1)}-1}{2} - 1} \exp\left\{-\frac{\lambda_i}{2}\sum_{j=1}^{n_{i(1)}}(y_{ij}-\hat{\mu}_i)^2\right\} \lambda_i^{\frac{1}{2}} \exp\left\{-\frac{n_{i(1)}\lambda_i}{2}(\mu_i - \mu_i^*)^2\right\} \quad (26)$$

This is $NG(\mu_{i,ni(1)}, k_{i,ni(1)}, \alpha_{i,ni(1)}, \beta_{i,ni(1)})$; $\mu_{i,n_{i(1)}} = \hat{\mu}_i$, $k_{i,ni(1)} = n_{i(1)}$, $\alpha_{i,ni(1)} = (n_{i(1)} - 1)/2$ and $\beta_{i,n_{i(1)}} = \frac{1}{2}\sum_{j=1}^{n_{i(1)}}(y_{ij} - \hat{\mu}_i)^2$. The marginal posterior of $\lambda_i$ becomes

$$\begin{aligned}
P(\lambda_i|y) \quad &\propto \lambda_i^{\alpha_{i,n_{i(1)}} - \frac{1}{2}} \exp\left\{-\lambda_i\beta_{i,n_{i(1)}}\right\} \int \exp\left\{-\frac{\lambda_i k_{i,n_{i(1)}}}{2}(\mu_i - \mu_{i,n_{i(1)}}^*)^2\right\} d\mu_i \\
&\propto \lambda_i^{\alpha_{i,n_{i(1)}} - \frac{1}{2}} \exp\left\{-\lambda_i\beta_{i,n_{i(1)}}\right\} \sqrt{\frac{2\pi}{\lambda_i k_{i,n_{i(1)}}}} \\
&\propto \lambda_i^{\alpha_{i,n_{i(1)}} - 1} \exp\left\{-\lambda_i\beta_{i,n_{i(1)}}\right\}
\end{aligned} \quad (27)$$

which is $\lambda_i|y \sim G(\lambda_i|\alpha_{i,ni(1)}, \beta_{i,ni(1)})$. This can be implied that $\sigma_i^2|y \sim IG(\sigma_i^2|\alpha_{i,ni(1)}, \beta_{i,ni(1)})$. Let $\rho_i = \beta_{i,n_{i(1)}} + \frac{k_{i,n_{i(1)}}}{2}(\mu - \mu_{i,n_{i(1)}})^2$ and $a_i = \lambda_i\rho_i$, then $d\lambda_i = \frac{1}{\rho_i}da_i$. Also, let $b_i = \alpha_{i,n_{i(1)}} + \frac{1}{2}$. From Eq. (26), the marginal posterior of $\mu_i|y$ is also obtained as

$$\begin{aligned}
P(\mu_i|\lambda_i, y)) \quad &\propto \int \lambda_i^{(\alpha_{i,n_{i(1)}} + \frac{1}{2}) - 1} \exp\left\{-\lambda_i\left[\beta_{i,n_{i(1)}} + \frac{k_{i,n_{i(1)}}}{2}(\mu - \mu_{i,n_{i(1)}})^2\right]\right\} d\lambda_i \\
&\propto \int \lambda_i^{b_i - 1} \exp\{-\lambda_i\rho_i\} d\lambda_i \\
&\propto \int \left(\frac{a_i}{\rho_i}\right)^{b_i - 1} \frac{1}{\rho_i} \exp(-a_i) da_i \\
&\propto \rho_i^{-b_i} \int a_i^{b_i - 1} \exp(-a_i) da_i \\
&\propto \rho_i^{-b_i} \\
&\propto \left[\beta_{i,n_{i(1)}} + \frac{k_{i,n_{i(1)}}}{2}(\mu_i - \mu_{i,n_{i(1)}})^2\right]^{-(\alpha_{i,n_{i(1)}} + \frac{1}{2})} \\
&\propto \left[1 + \frac{1}{2\alpha_{i,n_{i(1)}}}\frac{k_{i,n_{i(1)}}\alpha_{i,n_{i(1)}}}{\beta_{i,n_{i(1)}}}(\mu_i - \mu_{i,n_{i(1)}})^2\right]^{-(\alpha_{i,n_{i(1)}} + \frac{1}{2})}
\end{aligned} \quad (28)$$

Then, $\mu_i|y \sim t_{df}(\mu_i|\mu_{i,ni(1)}, \beta_{i,ni(1)}/[\alpha_{i,ni(1)}k_{i,ni(1)}])$ where df $= 2\alpha_{i,ni(1)} = n_{i(1)} - 1$. For $\delta_{i,ni(1)} = 1 - \delta_i$, *Jin, Thulin & Larsson (2017)* have investigated and attempted to find the power-divergence (PD) interval for $\delta_i$ estimated by Bayesian credible interval-based beta(d,d) prior. For focusing on this study, the beta(d,d) prior of $\delta_{i,ni(1)}$ is given by

$$P(\delta_{i,n_{i(1)}}) = \frac{\Gamma(2d_i)}{[\Gamma(d_i)]^2}\delta_{i,n_{i(1)}}^{d_i - 1}(1 - \delta_{i,n_{i(1)}})^{d_i - 1} \quad (29)$$

where $d_i = (2 + z_{\zeta/2}^2)/6$; $z_{\zeta/2}$ be a random sample of standard normal distribution. It is combined with the likelihood function of $\delta_{i,ni(1)}$ such that its posterior of $\delta_{i,ni(1)}$ is then

$$P(\delta_{i,n_{i(1)}}) \propto \frac{\Gamma(2d_i)}{[\Gamma(d_i)]^2}\delta_{i,n_{i(1)}}^{(n_{i(1)}+d_i)-1}(1 - \delta_{i,n_{i(1)}})^{(n_{i(0)}+d_i)-1} \quad (30)$$

---

**Algorithm 5.**

(1) Generate $X_{ij} \sim \Delta(\mu_i, \sigma^2_i, \delta_i)$; $i$ = 1, 2 and $j$ = 1, 2, ..., $n_i$.

(2) Compute $n_{i(0)}$, $n_{i(1)}$, $\hat{\mu}_i$, $\hat{\sigma}^2_i$ and $\hat{\delta}_i$.

(3) Construct CIs based on the methods as follows:

  • GCI, FGCI and MOVER from Algorithms 1, 2 and 3, respectively.

  • HPD-Jef, HPD-Rul and HPD-NG from Algorithm 4.

(4) Repeat 1–3, a number of times, (say, $M$ = 5,000). All CIs are obtained.

(5) Compute CPs and RALs with all CIs.

---

which is beta distribution, denoted as $beta(n_{i(1)} + d_i, n_{i(0)} + d_i)$. The posterior distributions of $\mu_i$ and $\sigma^2_i$ based on NG prior and $\delta_{i,ni(1)}$ based on *Jin, Thulin & Larsson (2017)* are obtained, then HPD-NG credible interval for $\theta$ can be computed in Algorithm 4.

## RESULTS

The CIs proposed in this study are HPD-NG and MOVER. The former develops the NG prior to obtain the posterior density, while the latter is an extension of Hasan and Krishnamoorthy's MOVER (*Hasan & Krishnamoorthy, 2018*). Both were compared with the existing CIs HPD-Jef and HPD-Rul of *Harvey & Van der Merwe (2012)*, GCI of *Wu & Hsieh (2014)*, and FGCI of *Hasan & Krishnamoorthy (2018)*. Two simulation studies were conducted to show the aforementioned CI performances under equal and unequal sample sizes in different situations:

S1. The probability of additional zero $\delta_i$ is varied while $\mu_i$ = 3 and $\sigma^2_i$ = 1.

S2. The mean $\mu_i$ and $\delta_i$ are varied while $\sigma^2_i$ = 1.

S1 was designed and simulated to be consistent with the weekly rainfall datasets, as can be seen in the next section. The second one was constructed to indicate CI performance when both the mean $\mu_i$ and $\delta_i$ are changed, i.e. whether the numerical simulation is consistent with S1.

The coverage probability (CP) and relative average length (RAL, denoted as the ratio of the average lengths of a proposed CI to HPD-Rul) were used to assess the performances of the CIs using Monte Carlo simulations. For 5,000 simulation runs, GPQs and FGPQs were fixed at 2,500 at a nominal level of 0.95. In a comparison of the methods, the following criteria were used to judge the best-performing CI: a CP closer to or greater than the nominal level and the narrowest RAL of less than 1 and minimal. The steps of the simulation procedure are given in Algorithm 5.

The findings are based on the simulation work as follows. For the ratio of variances, the numerical evaluation shows that for the method in the S1 scenario, MOVER provided good performance for a small difference between $\delta_i$ with equal small sample sizes (Table 1). Importantly, HPD-NG had good coverage (a CP greater than 0.95) and the shortest CIs for both equal and unequal medium sample sizes as well as a large difference in $\delta_i$ for unequal large sample sizes. Meanwhile, HPD-Rul's performance satisfied the criteria for a

small difference in $\delta_i$ and unequal large sample sizes. The results in Table 2 for case S2 indicate that for a small difference in $\delta_i$, MOVER could meet the requirements for equal small sample sizes, while HPD-NG maintained the given target for both equal and unequal medium sample sizes, and unequal large sample sizes. Moreover, HPD-Rul performed well for a large difference in $\delta_i$ and unequal medium-to-large sample sizes. From the evidence of both situations, HPD-Jef gave the best CP in all cases even though its average lengths were mostly wider than HPD-NG. Meanwhile, GCI and FGCI performances provided average lengths that were broader than other methods in all situations.

## An empirical application

*Duangdai & Likasiri (2017)* predicted the global temperature, and forest and seasonal rainfall amount in northern Thailand by various mathematical models. Their results indicate that during 1973–2008, the rainfall fluctuation during the rainy season was less than the summer rainfall, although the forest cover was higher in the summer than in the rainy season. Approximately 60% of the total arable land in the northeast has been turned over to rice cultivation (*Nicely & Counselor, 2018*). Importantly, both the northern and northeastern regions of Thailand are important agricultural areas where the planted rice is rain-fed and is the most valuable crop in the Lower Mekong Basin (*Zhang et al., 2014*). These findings led us to focus on the rainfall amounts in the northern and northeastern areas in Thailand, especially regarding rainfall variation in the rainy season. Importantly, a comparison of the rainfall in the northern and northeastern regions in terms of the ratio of variances was investigated and estimated with our proposed CIs. There were 272 substations in total for both regions to record rainfall measurements by the Thai Meteorological Center. The 272 rainfall observed values contained 40 of 62 (64.52%) and 145 of 210 (69.05%) positive records in the north (62 substations) and northeast (210 substations) areas during 2–8 July 2018, respectively. The remainder were zero observations for both sets.

To examine for normality, a Q–Q plot of the log-transformed positive rainfall amounts for the two areas are plotted in Fig. 1. Histograms also confirmed the fitted distributions of the northern and northeastern region rainfall records, as shown in Fig. 2. Moreover, Table 3 report the AIC results to check the fitted distribution of the positive rainfall observations for both areas. The results show that the positive rainfall for both had lognormal distributions, while the fact that the records contained zero values implies that both sets fit delta-lognormal distributions. The approximation of rainfall dispersion ratio between north and northeast areas is $\theta = -1.674$ [ratio = $\exp(-1.674) = 0.187$] where the variance $(\omega_1, \omega_2) = (448.34, 2{,}390.93)$; the basic statistics are $(n_1, n_2) = (62{,}210)$; $(\hat{\mu}_1, \hat{\mu}_2) = (2.541, 2.576)$; $(\hat{\sigma}_1^2, \hat{\sigma}_2^2) = (0.886, 1.576)$ and $(\hat{\delta}_1, \hat{\delta}_2) = (0.355, 0.309)$. From the results, the 95%CIs for $\exp(\theta)$ given in Table 4 indicate that HPD-NG and MOVER were better for situation S1 $[(\mu_1, \mu_2) = (3, 3); (\sigma_1^2, \sigma_2^2) = (1, 1)$ and $(\delta_1, \delta_2) = (0.3, 0.3)]$.

As mentioned previously, the results can be interpreted as the rainfall variability in the northern region being less than the northeastern one during 2–8 July 2018, which

**Table 1 CP and RAL performances of 95% CIs for θ: $(\mu_1, \mu_2) = (3, 3)$; $(\sigma_1^2, \sigma_2^2) = (1, 1)$.**

| $(n_1,n_2)$ | $(\delta_1,\delta_2)$ | CP | | | | | | RAL | | | | | |
|---|---|---|---|---|---|---|---|---|---|---|---|---|---|
| | | HPD-Jef | HPD-Rul | HPD-NG | GCI | FGCI | MOVER | HPD-Jef | HPD-Rul | HPD-NG | GCI | FGCI | MOVER |
| (15,15) | (0.1,0.1) | 99.8 | 99.7 | 99.8 | 99.7 | 99.6 | 94.4 | 1.095 | [*] | 1.152 | 1.109 | 1.108 | **0.796** |
| | (0.1,0.2) | 99.8 | 99.7 | 99.8 | 99.5 | 99.5 | 94.7 | 1.104 | [*] | 1.179 | 1.122 | 1.120 | **0.819** |
| | (0.1,0.3) | 99.8 | 99.4 | 99.8 | 99.3 | 99.3 | 94.0 | 1.115 | [*] | 1.206 | 1.135 | 1.134 | **0.847** |
| | (0.1,0.4) | 99.8 | 99.4 | 99.9 | 99.4 | 99.4 | 94.3 | 1.137 | [*] | 1.273 | 1.166 | 1.166 | **0.934** |
| | (0.1,0.5) | 99.9 | 99.6 | 99.9 | 99.6 | 99.6 | 95.5 | 1.160 | [*] | 1.344 | 1.201 | 1.200 | 1.045 |
| | (0.2,0.1) | 99.8 | 99.6 | 99.8 | 99.5 | 99.5 | 93.5 | 1.105 | [*] | 1.179 | 1.121 | 1.120 | – |
| | (0.2,0.2) | 99.9 | 99.6 | 99.9 | 99.6 | 99.6 | 94.0 | 1.114 | [*] | 1.205 | 1.131 | 1.130 | **0.836** |
| | (0.2,0.3) | 99.7 | 99.4 | 99.8 | 99.3 | 99.3 | 94.8 | 1.122 | [*] | 1.228 | 1.139 | 1.139 | **0.856** |
| | (0.2,0.4) | 99.9 | 99.5 | 99.9 | 99.5 | 99.4 | 94.3 | 1.146 | [*] | 1.296 | 1.171 | 1.171 | **0.943** |
| | (0.2,0.5) | 99.8 | 99.3 | 99.9 | 99.3 | 99.4 | 94.5 | 1.170 | [*] | 1.373 | 1.209 | 1.208 | 1.072 |
| | (0.3,0.1) | 99.8 | 99.5 | 99.8 | 99.4 | 99.4 | 94.2 | 1.114 | [*] | 1.204 | 1.133 | 1.131 | **0.845** |
| | (0.3,0.2) | 99.7 | 99.6 | 99.8 | 99.5 | 99.5 | 94.5 | 1.123 | [*] | 1.230 | 1.142 | 1.141 | **0.861** |
| | (0.3,0.3) | 99.8 | 99.2 | 99.8 | 99.2 | 99.2 | 94.2 | 1.131 | [*] | 1.253 | 1.151 | 1.149 | **0.877** |
| | (0.3,0.4) | 99.8 | 99.5 | 99.9 | 99.3 | 99.3 | 95.2 | 1.153 | [*] | 1.320 | 1.179 | 1.178 | **0.967** |
| | (0.3,0.5) | 99.9 | 99.5 | 99.9 | 99.5 | 99.5 | 95.8 | 1.177 | [*] | 1.393 | 1.210 | 1.210 | 1.070 |
| (50,50) | (0.1,0.1) | 97.3 | 97.2 | 95.5 | 96.9 | 97.0 | 91.2 | 1.019 | [*] | **0.935** | 1.027 | 1.026 | – |
| | (0.1,0.2) | 97.6 | 97.2 | 95.4 | 97.2 | 97.2 | 90.7 | 1.021 | [*] | **0.936** | 1.029 | 1.028 | – |
| | (0.1,0.3) | 96.9 | 96.5 | 94.6 | 96.7 | 96.5 | 90.6 | 1.022 | [*] | **0.939** | 1.031 | 1.030 | – |
| | (0.1,0.4) | 97.2 | 96.7 | 94.8 | 96.5 | 96.7 | 89.9 | 1.024 | [*] | **0.944** | 1.034 | 1.034 | – |
| | (0.1,0.5) | 96.8 | 95.9 | 94.4 | 95.8 | 96.1 | 89.4 | 1.029 | [*] | **0.955** | 1.041 | 1.041 | – |
| | (0.2,0.1) | 97.2 | 96.6 | 94.9 | 96.6 | 96.6 | 90.5 | 1.020 | [*] | **0.936** | 1.028 | 1.027 | – |
| | (0.2,0.2) | 97.4 | 96.8 | 95.1 | 96.6 | 96.7 | 90.4 | 1.022 | [*] | **0.938** | 1.029 | 1.029 | – |
| | (0.2,0.3) | 96.4 | 96.1 | 94.1 | 95.8 | 95.9 | 89.2 | 1.023 | [*] | **0.941** | 1.032 | 1.032 | – |
| | (0.2,0.4) | 96.8 | 96.3 | 94.4 | 96.1 | 96.0 | 90.2 | 1.026 | [*] | **0.946** | 1.035 | 1.035 | – |
| | (0.2,0.5) | 96.8 | 96.5 | 95.1 | 96.4 | 96.5 | 90.1 | 1.030 | [*] | **0.956** | 1.041 | 1.041 | – |
| | (0.3,0.1) | 97.4 | 96.9 | 95.4 | 96.8 | 96.9 | 90.6 | 1.022 | [*] | **0.940** | 1.031 | 1.030 | – |
| | (0.3,0.2) | 96.9 | 96.5 | 94.5 | 96.5 | 96.3 | 88.5 | 1.024 | [*] | **0.942** | 1.032 | 1.032 | – |
| | (0.3,0.3) | 96.6 | 96.0 | 94.1 | 95.8 | 95.9 | 89.0 | 1.025 | [*] | **0.945** | 1.034 | 1.034 | – |
| | (0.3,0.4) | 97.1 | 96.6 | 95.2 | 96.8 | 96.7 | 89.9 | 1.027 | [*] | **0.950** | 1.036 | 1.035 | – |
| | (0.3,0.5) | 96.9 | 96.4 | 94.7 | 96.1 | 96.1 | 89.7 | 1.032 | [*] | **0.960** | 1.042 | 1.042 | – |
| (30,50) | (0.1,0.1) | 98.1 | 97.8 | 96.8 | 97.7 | 97.8 | 90.8 | 1.030 | [*] | **0.964** | 1.042 | 1.042 | – |
| | (0.1,0.2) | 98.3 | 97.9 | 96.7 | 97.8 | 97.7 | 90.6 | 1.030 | [*] | **0.966** | 1.043 | 1.042 | – |
| | (0.1,0.3) | 97.8 | 97.7 | 96.2 | 97.1 | 97.1 | 89.3 | 1.032 | [*] | **0.968** | 1.044 | 1.042 | – |
| | (0.1,0.4) | 97.7 | 97.4 | 96.0 | 97.1 | 97.1 | 90.1 | 1.033 | [*] | **0.972** | 1.043 | 1.042 | – |
| | (0.1,0.5) | 97.6 | 97.2 | 96.3 | 97.0 | 97.1 | 89.7 | 1.036 | [*] | **0.981** | 1.046 | 1.046 | – |
| | (0.2,0.1) | 98.0 | 97.7 | 96.6 | 97.4 | 97.5 | 90.9 | 1.033 | [*] | **0.971** | 1.047 | 1.046 | – |
| | (0.2,0.2) | 97.8 | 97.4 | 96.3 | 97.3 | 97.3 | 90.2 | 1.034 | [*] | **0.972** | 1.047 | 1.046 | – |
| | (0.2,0.3) | 97.9 | 97.4 | 96.4 | 97.2 | 97.2 | 89.9 | 1.034 | [*] | **0.974** | 1.046 | 1.046 | – |
| | (0.2,0.4) | 98.1 | 97.6 | 96.4 | 97.3 | 97.3 | 89.6 | 1.035 | [*] | **0.978** | 1.047 | 1.046 | – |
| | (0.2,0.5) | 97.6 | 97.5 | 96.2 | 97.0 | 96.9 | 89.5 | 1.039 | [*] | **0.987** | 1.050 | 1.049 | – |
| | (0.3,0.1) | 98.1 | 97.6 | 96.4 | 97.3 | 97.3 | 89.8 | 1.036 | [*] | **0.981** | 1.054 | 1.053 | – |

| $(n_1,n_2)$ | $(\delta_1,\delta_2)$ | CP | | | | | | RAL | | | | | |
|---|---|---|---|---|---|---|---|---|---|---|---|---|---|
| | | HPD-Jef | HPD-Rul | HPD-NG | GCI | FGCI | MOVER | HPD-Jef | HPD-Rul | HPD-NG | GCI | FGCI | MOVER |
| | (0.3,0.2) | 97.7 | 97.1 | 96.0 | 96.7 | 96.9 | 89.6 | 1.037 | [*] | **0.981** | 1.053 | 1.053 | – |
| | (0.3,0.3) | 97.6 | 97.1 | 95.6 | 96.7 | 96.8 | 89.6 | 1.038 | [*] | **0.984** | 1.053 | 1.052 | – |
| | (0.3,0.4) | 97.4 | 97.0 | 95.8 | 96.7 | 96.7 | 88.8 | 1.039 | [*] | **0.988** | 1.053 | 1.052 | – |
| | (0.3,0.5) | 97.7 | 97.3 | 96.2 | 97.1 | 97.0 | 89.6 | 1.043 | [*] | **0.997** | 1.056 | 1.055 | – |
| (50,100) | (0.1,0.1) | 96.7 | 96.4 | 93.9 | 96.2 | 96.3 | 91.7 | 1.016 | [*] | 0.922 | 1.026 | 1.026 | – |
| | (0.1,0.2) | 96.6 | 96.2 | 94.0 | 96.0 | 96.2 | 91.4 | 1.016 | [*] | 0.921 | 1.025 | 1.025 | – |
| | (0.1,0.3) | 96.7 | 96.2 | 93.8 | 96.1 | 96.1 | 91.1 | 1.016 | [*] | 0.921 | 1.025 | 1.025 | – |
| | (0.1,0.4) | 96.7 | 96.6 | 94.2 | 96.3 | 96.4 | 91.3 | 1.016 | [*] | **0.921** | 1.025 | 1.025 | – |
| | (0.1,0.5) | 96.8 | 96.5 | 94.5 | 96.3 | 96.3 | 90.9 | 1.019 | [*] | **0.923** | 1.026 | 1.025 | – |
| | (0.2,0.1) | 96.7 | 96.4 | 94.1 | 96.1 | 96.3 | 91.1 | 1.017 | [*] | **0.923** | 1.028 | 1.028 | – |
| | (0.2,0.2) | 96.7 | 96.3 | 93.7 | 95.9 | 96.0 | 90.4 | 1.017 | [*] | 0.923 | 1.029 | 1.027 | – |
| | (0.2,0.3) | 96.4 | 96.1 | 93.9 | 96.0 | 95.9 | 90.5 | 1.017 | [*] | 0.922 | 1.028 | 1.027 | – |
| | (0.2,0.4) | 96.8 | 96.6 | 93.7 | 96.4 | 96.4 | 90.7 | 1.018 | [*] | 0.923 | 1.028 | 1.027 | – |
| | (0.2,0.5) | 96.4 | 96.1 | 93.7 | 96.1 | 96.1 | 90.2 | 1.018 | [*] | 0.924 | 1.028 | 1.027 | – |
| | (0.3,0.1) | 96.0 | 96.0 | 93.0 | 95.5 | 95.4 | 90.9 | 1.019 | [*] | 0.926 | 1.033 | 1.033 | – |
| | (0.3,0.2) | 96.0 | 95.7 | 93.0 | 95.3 | 95.4 | 90.2 | 1.019 | [*] | 0.926 | 1.032 | 1.031 | – |
| | (0.3,0.3) | 96.4 | 95.7 | 93.7 | 95.8 | 95.8 | 90.5 | 1.019 | [*] | 0.926 | 1.032 | 1.031 | – |
| | (0.3,0.4) | 96.4 | 95.7 | 93.4 | 95.9 | 95.9 | 90.6 | 1.020 | [*] | 0.926 | 1.031 | 1.031 | – |
| | (0.3,0.5) | 96.7 | 96.3 | 94.2 | 96.2 | 96.3 | 90.7 | 1.021 | [*] | 0.928 | 1.030 | 1.031 | – |

**Notes:**
[*]: HPD-Rul satisfies the criteria.
Bold denotes the best-performing CI.

implies that growing rice in the northeastern region was probably more affected than in the northern region because the former's rain variation was larger. It is possible that this information could influence approximately 60% of the total arable area in the northeast. Note that *Nicely & Counselor (2018)* estimated 70% of rice production during the marketing year 2018–2019 cultivated under desirable weather conditions. This analysis might provide useful information to the Royal Thai Government to carry out effective water management. Our findings made a few realizations about natural disasters due to climate change during the rainy season in the northern and northeastern Thailand as well. The results are also in agreement with the simulation study ones in Table 1.

## DISCUSSION

The HPD-NG was proposed for constructing confidence intervals for comparing the rainfall dispersion in north and northeast regions. How to select the prior in this situation is that we found the CP performance of HPD depend on the posterior densities of $\sigma^2$ and $\delta$ in the previous study. To our knowledge, we believed that $\mu$ and $\sigma^2$ might be suitable for random variables of the normal and gamma distributions, respectively. This becomes the normal-gamma prior of $(\mu,\sigma^2)$. For $\delta$, it was motivated by Jeffreys' prior, while a beta distribution was also developed and recommended by *Jin, Thulin & Larsson (2017)* so that beta $(d,d)$ becomes a prior of $\delta$ in this study.

**Table 2 CP and RAL performances of 95% CIs for θ: $(\sigma_2^1, \sigma_2^2) = (1, 1)$.**

| $(n_1,n_2)$ | $(\delta_1,\delta_2)$ | $(\mu_1,\mu_2)$ | CP | | | | | | RAL | | | | | |
|---|---|---|---|---|---|---|---|---|---|---|---|---|---|---|
| | | | HPD-Jef | HPD-Rul | HPD-NG | GCI | FGCI | MOVER | HPD-Jef | HPD-Rul | HPD-NG | GCI | FGCI | MOVER |
| (15,15) | (0.1,0.1) | (0,0) | 99.8 | 99.5 | 99.8 | 99.5 | 99.5 | 93.7 | 1.096 | [*] | 1.154 | 1.112 | 1.110 | 0.797 |
| | | (0,0.3) | 99.8 | 99.4 | 99.6 | 99.4 | 99.3 | 93.7 | 1.096 | [*] | 1.155 | 1.111 | 1.108 | 0.798 |
| | | (0,0.5) | 99.9 | 99.6 | 99.9 | 99.7 | 99.7 | 93.9 | 1.095 | [*] | 1.153 | 1.111 | 1.108 | 0.796 |
| | | (0,0.7) | 99.7 | 99.5 | 99.7 | 99.6 | 99.6 | 94.4 | 1.095 | [*] | 1.153 | 1.111 | 1.109 | 0.798 |
| | | (0,0.9) | 99.8 | 99.6 | 99.7 | 99.4 | 99.5 | 93.8 | 1.096 | [*] | 1.154 | 1.111 | 1.110 | 0.797 |
| | (0.2,0.2) | (0,0) | 99.7 | 99.4 | 99.8 | 99.4 | 99.4 | 93.9 | 1.114 | [*] | 1.205 | 1.132 | 1.130 | **0.839** |
| | | (0,0.3) | 99.9 | 99.6 | 99.9 | 99.6 | 99.7 | 94.6 | 1.113 | [*] | 1.205 | 1.130 | 1.129 | **0.835** |
| | | (0,0.5) | 99.9 | 99.5 | 99.8 | 99.5 | 99.6 | 94.5 | 1.115 | [*] | 1.206 | 1.131 | 1.129 | **0.835** |
| | | (0,0.7) | 99.8 | 99.7 | 99.9 | 99.6 | 99.5 | 94.6 | 1.114 | [*] | 1.205 | 1.132 | 1.130 | **0.838** |
| | | (0,0.9) | 99.8 | 99.6 | 99.9 | 99.5 | 99.6 | 94.5 | 1.114 | [*] | 1.205 | 1.131 | 1.130 | **0.836** |
| | (0.4,0.4) | (0,0) | 99.7 | 99.2 | 99.9 | 99.2 | 99.3 | 94.6 | 1.174 | [*] | 1.382 | 1.200 | 1.199 | 1.038 |
| | | (0,0.3) | 99.8 | 99.4 | 99.9 | 99.2 | 99.2 | 94.7 | 1.173 | [*] | 1.380 | 1.201 | 1.199 | 1.033 |
| | | (0,0.5) | 99.9 | 99.5 | 99.9 | 99.5 | 99.5 | 95.2 | 1.175 | [*] | 1.384 | 1.202 | 1.200 | 1.033 |
| | | (0,0.7) | 99.9 | 99.4 | 99.9 | 99.5 | 99.5 | 95.1 | 1.174 | [*] | 1.383 | 1.198 | 1.199 | 1.029 |
| | | (0,0.9) | 99.9 | 99.5 | 99.9 | 99.5 | 99.5 | 95.5 | 1.174 | [*] | 1.383 | 1.201 | 1.199 | 1.038 |
| (30,30) | (0.1,0.1) | (0,0) | 98.8 | 98.3 | 97.7 | 98.3 | 98.4 | 90.8 | 1.037 | [*] | **0.986** | 1.046 | 1.044 | – |
| | | (0,0.3) | 98.8 | 98.4 | 97.7 | 98.3 | 98.2 | 90.8 | 1.037 | [*] | **0.986** | 1.047 | 1.045 | – |
| | | (0,0.5) | 98.9 | 98.6 | 97.9 | 98.3 | 98.4 | 91.0 | 1.037 | [*] | **0.986** | 1.047 | 1.045 | – |
| | | (0,0.7) | 98.7 | 98.2 | 97.7 | 98.1 | 98.0 | 90.5 | 1.037 | [*] | **0.987** | 1.047 | 1.045 | – |
| | | (0,0.9) | 98.8 | 98.4 | 98.0 | 98.5 | 98.4 | 91.6 | 1.036 | [*] | **0.986** | 1.046 | 1.045 | – |
| | (0.2,0.2) | (0,0) | 98.4 | 98.2 | 97.7 | 98.2 | 98.2 | 90.3 | 1.041 | [*] | **0.997** | 1.052 | 1.051 | – |
| | | (0,0.3) | 98.4 | 98.2 | 97.6 | 98.1 | 98.0 | 90.3 | 1.041 | [*] | **0.997** | 1.052 | 1.051 | – |
| | | (0,0.5) | 98.2 | 97.8 | 97.3 | 97.7 | 97.6 | 90.0 | 1.042 | [*] | **0.997** | 1.052 | 1.050 | – |
| | | (0,0.7) | 98.4 | 98.0 | 97.5 | 97.7 | 97.8 | 89.9 | 1.040 | [*] | **0.998** | 1.051 | 1.051 | – |
| | | (0,0.9) | 98.1 | 98.0 | 97.4 | 97.7 | 97.8 | 89.9 | 1.041 | [*] | **0.998** | 1.051 | 1.051 | – |
| | (0.4,0.4) | (0,0) | 98.1 | 97.8 | 97.3 | 97.4 | 97.3 | 89.6 | 1.058 | [*] | 1.042 | 1.070 | 1.071 | – |
| | | (0,0.3) | 98.2 | 97.8 | 97.6 | 97.7 | 97.7 | 90.3 | 1.060 | [*] | 1.042 | 1.072 | 1.072 | – |
| | | (0,0.5) | 98.2 | 97.7 | 97.3 | 97.4 | 97.4 | 89.9 | 1.058 | [*] | 1.042 | 1.072 | 1.071 | – |
| | | (0,0.7) | 98.4 | 97.7 | 97.4 | 97.5 | 97.5 | 89.5 | 1.059 | [*] | 1.041 | 1.071 | 1.070 | – |
| | | (0,0.9) | 98.2 | 97.7 | 97.2 | 97.4 | 97.3 | 89.8 | 1.058 | [*] | 1.041 | 1.070 | 1.071 | – |
| (50,50) | (0.1,0.1) | (0,0) | 97.1 | 96.8 | 95.1 | 96.7 | 96.8 | 90.1 | 1.020 | [*] | **0.936** | 1.028 | 1.027 | – |
| | | (0,0.3) | 97.3 | 96.9 | 95.3 | 97.0 | 96.8 | 91.0 | 1.020 | [*] | **0.935** | 1.028 | 1.027 | – |
| | | (0,0.5) | 97.7 | 97.2 | 95.3 | 97.1 | 97.1 | 90.5 | 1.019 | [*] | **0.934** | 1.027 | 1.026 | – |
| | | (0,0.7) | 97.4 | 97.1 | 95.2 | 97.0 | 96.9 | 90.9 | 1.019 | [*] | **0.935** | 1.027 | 1.026 | – |
| | | (0,0.9) | 96.8 | 96.4 | 94.8 | 96.2 | 96.3 | 89.9 | 1.020 | [*] | **0.936** | 1.028 | 1.027 | – |
| | (0.2,0.2) | (0,0) | 96.4 | 96.0 | 94.3 | 96.1 | 96.0 | 89.5 | 1.021 | [*] | **0.938** | 1.029 | 1.028 | – |
| | | (0,0.3) | 97.1 | 96.7 | 94.6 | 96.5 | 96.6 | 89.8 | 1.022 | [*] | **0.938** | 1.031 | 1.030 | – |
| | | (0,0.5) | 96.8 | 96.4 | 94.4 | 96.2 | 96.2 | 89.8 | 1.022 | [*] | **0.938** | 1.030 | 1.029 | – |
| | | (0,0.7) | 96.9 | 96.5 | 94.5 | 96.3 | 96.3 | 90.3 | 1.023 | [*] | **0.938** | 1.030 | 1.029 | – |
| | | (0,0.9) | 96.8 | 96.4 | 94.5 | 96.2 | 96.2 | 89.9 | 1.022 | [*] | **0.937** | 1.029 | 1.029 | – |

| $(n_1,n_2)$ | $(\delta_1,\delta_2)$ | $(\mu_1,\mu_2)$ | CP | | | | | | RAL | | | | | |
|---|---|---|---|---|---|---|---|---|---|---|---|---|---|---|
| | | | HPD-Jef | HPD-Rul | HPD-NG | GCI | FGCI | MOVER | HPD-Jef | HPD-Rul | HPD-NG | GCI | FGCI | MOVER |
| | (0.4,0.4) | (0,0) | 96.6 | 96.3 | 94.4 | 96.0 | 95.9 | 89.1 | 1.029 | [*] | **0.954** | 1.038 | 1.038 | – |
| | | (0,0.3) | 96.8 | 96.4 | 94.4 | 96.0 | 96.1 | 88.9 | 1.029 | [*] | **0.954** | 1.038 | 1.038 | – |
| | | (0,0.5) | 96.9 | 96.4 | 94.6 | 95.9 | 96.2 | 89.3 | 1.029 | [*] | **0.954** | 1.038 | 1.038 | – |
| | | (0,0.7) | 96.7 | 96.0 | 94.3 | 95.7 | 95.8 | 89.2 | 1.029 | [*] | **0.955** | 1.038 | 1.038 | – |
| | | (0,0.9) | 97.0 | 96.3 | 94.7 | 96.1 | 96.1 | 89.2 | 1.030 | [*] | **0.955** | 1.039 | 1.038 | – |
| (30,50) | (0.1,0.1) | (0,0) | 98.4 | 98.0 | 96.8 | 97.8 | 97.7 | 90.8 | 1.029 | [*] | **0.965** | 1.042 | 1.041 | – |
| | | (0,0.3) | 98.3 | 98.1 | 97.0 | 97.8 | 97.9 | 90.5 | 1.029 | [*] | **0.965** | 1.042 | 1.041 | – |
| | | (0,0.5) | 98.5 | 98.1 | 97.0 | 97.7 | 97.7 | 91.2 | 1.030 | [*] | **0.965** | 1.043 | 1.041 | – |
| | | (0,0.7) | 98.2 | 97.8 | 96.5 | 97.5 | 97.4 | 89.9 | 1.029 | [*] | **0.964** | 1.042 | 1.040 | – |
| | | (0,0.9) | 98.1 | 97.8 | 96.7 | 97.7 | 97.7 | 90.0 | 1.030 | [*] | **0.964** | 1.042 | 1.041 | – |
| | (0.2,0.2) | (0,0) | 98.1 | 97.7 | 96.7 | 97.5 | 97.5 | 90.1 | 1.033 | [*] | **0.972** | 1.048 | 1.046 | – |
| | | (0,0.3) | 97.8 | 97.5 | 96.3 | 97.3 | 97.3 | 90.2 | 1.033 | [*] | **0.972** | 1.047 | 1.046 | – |
| | | (0,0.5) | 98.1 | 97.8 | 96.8 | 97.4 | 97.4 | 90.5 | 1.033 | [*] | **0.972** | 1.047 | 1.046 | – |
| | | (0,0.7) | 98.1 | 97.6 | 96.3 | 97.3 | 97.3 | 90.1 | 1.033 | [*] | **0.971** | 1.046 | 1.045 | – |
| | | (0,0.9) | 97.8 | 97.2 | 96.0 | 97.0 | 97.0 | 89.6 | 1.034 | [*] | **0.972** | 1.047 | 1.047 | – |
| | (0.4,0.4) | (0,0) | 97.4 | 97.1 | 96.3 | 96.7 | 96.9 | 89.6 | 1.046 | [*] | 1.003 | 1.063 | 1.062 | – |
| | | (0,0.3) | 97.6 | 97.3 | 96.4 | 96.8 | 96.8 | 89.7 | 1.046 | [*] | 1.004 | 1.063 | 1.062 | – |
| | | (0,0.5) | 98.2 | 97.4 | 96.6 | 97.2 | 97.1 | 89.5 | 1.046 | [*] | 1.003 | 1.062 | 1.062 | – |
| | | (0,0.7) | 97.5 | 97.1 | 95.9 | 96.6 | 96.6 | 88.7 | 1.046 | [*] | 1.004 | 1.063 | 1.062 | – |
| | | (0,0.9) | 97.6 | 96.9 | 96.0 | 96.6 | 96.7 | 88.5 | 1.046 | [*] | 1.004 | 1.063 | 1.063 | – |
| (50,100) | (0.1,0.1) | (0,0) | 96.7 | 96.3 | 94.1 | 96.4 | 96.2 | 91.6 | 1.016 | [*] | **0.921** | 1.026 | 1.025 | – |
| | | (0,0.3) | 96.7 | 96.5 | 94.3 | 96.3 | 96.3 | 92.3 | 1.016 | [*] | **0.922** | 1.026 | 1.025 | – |
| | | (0,0.5) | 96.7 | 96.5 | 93.9 | 96.3 | 96.1 | 92.0 | 1.015 | [*] | **0.921** | 1.026 | 1.025 | – |
| | | (0,0.7) | 96.5 | 96.3 | 93.8 | 96.0 | 96.0 | 92.0 | 1.016 | [*] | **0.922** | 1.026 | 1.025 | – |
| | | (0,0.9) | 97.1 | 96.7 | 94.2 | 96.6 | 96.5 | 92.1 | 1.016 | [*] | **0.921** | 1.025 | 1.025 | – |
| | (0.2,0.2) | (0,0) | 96.0 | 95.5 | 93.2 | 95.5 | 95.4 | 90.4 | 1.017 | [*] | 0.923 | 1.029 | 1.028 | – |
| | | (0,0.3) | 96.7 | 96.5 | 94.0 | 96.5 | 96.4 | 91.3 | 1.017 | [*] | 0.922 | 1.029 | 1.028 | – |
| | | (0,0.5) | 96.3 | 95.9 | 93.5 | 95.8 | 95.7 | 90.8 | 1.018 | [*] | 0.923 | 1.029 | 1.028 | – |
| | | (0,0.7) | 96.6 | 96.3 | 93.8 | 96.1 | 96.1 | 91.0 | 1.016 | [*] | 0.923 | 1.028 | 1.027 | – |
| | | (0,0.9) | 96.7 | 96.1 | 93.7 | 96.2 | 96.2 | 91.2 | 1.018 | [*] | 0.923 | 1.029 | 1.028 | – |
| | (0.4,0.4) | (0,0) | 96.1 | 95.5 | 93.2 | 95.4 | 95.4 | 89.5 | 1.023 | [*] | 0.932 | 1.035 | 1.035 | – |
| | | (0,0.3) | 96.4 | 95.8 | 93.0 | 95.8 | 95.7 | 89.3 | 1.023 | [*] | 0.933 | 1.036 | 1.036 | – |
| | | (0,0.5) | 96.3 | 95.6 | 93.2 | 95.7 | 95.6 | 89.3 | 1.023 | [*] | 0.933 | 1.036 | 1.036 | – |
| | | (0,0.7) | 96.3 | 96.0 | 93.6 | 96.0 | 95.9 | 90.1 | 1.023 | [*] | 0.932 | 1.035 | 1.036 | – |
| | | (0,0.9) | 96.2 | 95.8 | 93.7 | 95.8 | 95.9 | 89.4 | 1.024 | [*] | 0.933 | 1.036 | 1.037 | – |

**Notes:**
[*]: HPD-Rul satisfies the criteria.
Bold denotes the best-performing CI.

The difference between HPD-Jef and HPD-Rul is the posterior densities of $\sigma^2$ and $\delta$, although it can be seen from the results of the simulation study that HPD-Rul's outcomes were agreement with *Harvey & Van der Merwe (2012)* when focusing on the ratio of

**Table 3 Results of AIC for the nonzero rainfall records in north and northeast areas.**

| Regions | AIC | | | | | | |
|---|---|---|---|---|---|---|---|
| | **Exponential** | **Weibull** | **Lognormal** | **Normal** | **t-distribution** | **Cauchy** | **Logistic** |
| Northern | 316.677 | 317.046 | **314.927** | 348.812 | 332.373 | 336.093 | 337.678 |
| Northeastern | 1,232.797 | 1,231.199 | **1,227.468** | 1,411.599 | 1,332.576 | 1,331.557 | 1,376.586 |

Note:
Bold denotes the lowest AIC.

**Table 4 95% CIs for the weekly rainfall ratio between north and northeast regions.**

| Methods | 95% CIs for exp ($\theta$) | | Length |
|---|---|---|---|
| | **Lower** | **Upper** | |
| HPD-Jef | 0.0455 | 0.9255 | 0.8800 |
| HPD-Rul | 0.0477 | 0.9239 | 0.8762 |
| HPD-NG | 0.0546 | 0.7793 | 0.7247 |
| GCI | 0.0464 | 0.9860 | 0.9396 |
| FGCI | 0.0652 | 1.1484 | 1.0832 |
| MOVER | 0.0512 | 0.5896 | 0.5384 |

delta-lognormal means. This implies that both HPD performances are dependent on the posterior of $\sigma^2$ and $\delta$. For the proposed HPD-NG, its prior of $\tau = (\mu, \lambda)$; $\lambda = 1/\sigma^2$ is the inverse of the Jeffreys' prior, leading us to obtain the posterior distributions of $\mu$ and $\sigma^2$ derived from the NG prior under the assumption that the mean $\mu$ has a normal distribution and the precision $\lambda$ has a gamma distribution. After that, the posterior of $\sigma^2$ has a inverse-gamma distribution with its parameters $(\alpha_{i,ni(1)}, \beta_{i,ni(1)})$. Importantly, the parameter $\beta_{i,ni(1)}$ is different from the inverse-gamma based one on Jeffreys' prior, resulting in a different posterior of $\mu$ as well. Likewise, the prior of $\delta$ was found and recommended by *Jin, Thulin & Larsson (2017)*. For these reasons, the HPD-NG prior was developed to estimate the ratio between delta-lognormal variances. However, more research on this topic needs to be conduct to find an appropriate prior to obtain a better performance. Furthermore, MOVER could maintain its performance to satisfy the target criteria even with a small difference between the binomial proportions and small sample sizes. It is possible that an interval estimate for $\delta$ based on Wilson's interval satisfies the criteria for small to moderate sample sizes, as confirmed by *Donner & Zou (2011)*.

## CONCLUSIONS

The purpose of this study was to develop CIs, namely HPD-NG and MOVER, for the ratio of delta-lognormal variances through Monte Carlo simulations. By way of comparison, both were examined to report their performance with the existing methods: HPD-Jef, HPD-Rul, GCI and FGCI. These proposed CIs were applied to estimate and compare the rainfall fluctuation between the northern and northeastern regions in Thailand in terms of the ratio of variances.

The findings of this study indicate that HPD-NG is the best and most recommended CI in situations S1 (with varied $\delta$) for both equal and unequal medium sample sizes, and for a

large difference between the binomial proportions for unequal large sample sizes. Both MOVER and HPD-Rul are the next best CIs recommended for a small difference between the binomial proportions with different sample sizes: MOVER for equal small sample sizes and HPD-Rul for unequal large sample sizes. For situation S2 where $\delta$ and $\mu$ are both varied, HPD-NG delivered the best performance for a small binomial proportion and both equal and unequal medium sample sizes as well as a small proportion of zeros for unequal large sample sizes. MOVER is also recommended for situations similar to situation S1, while HPD-Rul is recommended for a large proportion of zeros and unequal medium-to-large sample sizes. Although the CPs of HPD-Jef, GCI and FGCI results satisfied the criteria, their average lengths were wider than our proposed methods in all situations.

Note: In this paper, confidence intervals for the ratio variances of delta-lognormal are proposed, whereas, other submission proposed CIs for the difference between variances of delta-lognormal populations based on *Herbert et al. (2011)* and *Krishnamoorthy, Lian & Mondal (2010)*. Although, the difference between them is that the former was considered to estimate the ratio of two delta-lognormal variances $(\omega_1/\omega_2)$ using HPD-based normal gamma prior. The latter was presented HPDs based on other priors for the difference in delta-lognormal variances $(\omega_1 - \omega_2)$ for large variational situations. According to *Herbert et al. (2011)* and *Krishnamoorthy, Lian & Mondal (2010)*, the assessment of the range of distribution of $(\omega_1 - \omega_2)$ can be estimated using CIs for $(\omega_1 - \omega_2)$ such that it can be implied that there is the distributed range between $(\omega_1 - \omega_2)$ and $(\omega_1/\omega_2)$. Finally, both of random variables are used to check significantly difference in the two independent datasets. The next contrast between this submission and other one is the illustrated data that there are different dispersions and locations in each paper.

### Funding
This research was funded by King Mongkut's University of Technology North Bangkok (Grant No. KMUTNB-62-KNOW-19). The first author was funded by Thailand Science Research and Innovation and National Research Council of Thailand: Contract no. PHD/0198/2561. This article was also supported by the Department of Computer Engineering, Khon Kaen University for GPU computing server. The funders had no role in study design, data collection and analysis, decision to publish, or preparation of the manuscript.

### Grant Disclosures
The following grant information was disclosed by the authors:
King Mongkut's University of Technology North Bangkok: KMUTNB-62-KNOW-19.
Thailand Science Research and Innovation and National Research Council of Thailand: PHD/0198/2561.

### Competing Interests
The authors declare that they have no competing interests.

## Author Contributions

- Patcharee Maneerat performed the experiments, analyzed the data, prepared figures and/or tables, authored or reviewed drafts of the paper, and approved the final draft.
- Sa-aat Niwitpong conceived and designed the experiments, analyzed the data, authored or reviewed drafts of the paper, and approved the final draft.
- Suparat Niwitpong conceived and designed the experiments, prepared figures and/or tables, authored or reviewed drafts of the paper, find data set, and approved the final draft.

## Data Availability

R code and data are available in the Supplemental Files.

## Supplemental Information

Supplemental information for this article can be found online at http://dx.doi.org/10.7717/peerj.8502#supplemental-information.

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
