# Peer review of "A Bayesian approach to construct confidence intervals for comparing the rainfall dispersion in Thailand"

_PeerJ, doi:10.7717/peerj.8502_

## Round 0.1 · original submission · Major Revisions

Dear authors,

Two reviewers suggest minor revisions to the manuscript and another one to reject it. A suggestion for a more descriptive title should be taken into consideration. Also, one reviewer suggests a clearer explanation of the statistical procedures as the technical language is difficult to follow for a broader audience.

The reviewer number three suggests that the paper should be rejected based on the lack of novelty of the paper, as its contribution is not clear. The paper does not offer a new methodology in the field.

Improving explanation of the methodology for a broader audience is a must. A clear explanation of the novel contribution of the article in the context of previous work must be added to the paper.

I suggest miajor revisions.

Reviewer 1 ·

Basic reporting

The contents of this paper are very interested and suitable for publishing in this journal. There are some minor mistake for improving as follows:

1.The title is not suitable. It is not cover all details of the paper.

2.The author should show the source code of all algorithm (If possible).

3.In HPD credible interval, there are many prior distribution, i.e. Jeffreys’ prior, Jeffreys’ Rule prior, Normal-gamma prior. How do the researchers select the prior distribution in real situation? The author should mention in this matter.

4. Which is the program for the simulation?

Experimental design

The experimental design is suitable. Which is the program for the simulation?

Validity of the findings

The findings for the paper are value to academic.

Reviewer 2 ·

Basic reporting

I think the paper is fine. I find it at stages a bit technical and it might be helpful to write in wording what is actually going on.

Experimental design

The data are coming from routinely collected data and this should be fine.

Validity of the findings

The findings make senes to me but it should be clearly said this is based on the simulation work, which can always be only limitied.

Additional comments

I think the paper is fine. At stages, the text is very technical and it might be better to have a lighter wording for the journal of this type.

It should be pointed out how this paper is different from papers in submission with PerrJ already or with Biometrical Journal or other relevant submissions as we do not want to see the same work publsihed twice.

Details:
p2, line 69: "guaranteed" should be "states" or "argue"
p2, line 89: "Methods" should be "methods"
p4, 126: "be p-dimension" should be " be a p-dimensional"
p15, 368: incomplete reference
p16, 407 delete "Biometrische Zeitschrift"
Lower Mekong Basin: use consistetn spelling throughout

Reviewer 3 ·

Basic reporting

The English is good with few minor typos (e.g.: line 84 stands should be stand). But the structure of the paper and its flow is somewhat weak. For example, Tables were not called in order in the text. e.g.: Table 3 (in the text) was called before Table 2 and 1, and Table 4 after table 5, which is very confusing.

Contribution of the paper is marginal and conclusions were not backed well by the analysis in the paper (see below). Overall, It is not clear what authors try to show or contribute in broad.

Experimental design

The purpose of this study was to develop CIs for the ratio of delta lognormal variances through Monte Carlo simulations. Building intervals is quite straightforward. I did not see any significant methodological contributions in the paper.

Validity of the findings

Conclusions are not solid. For example: in the conclusion section, authors noted that "The findings of this study indicate that HPD-NG is undoubtedly the best and most recommended CI (with varied δ) for both equal and unequal medium sample sizes" but how? and why it is "undoubtedly" the best? just by evaluating an empirical example?

---

## Round 0.2 · accepted · Accept

The reviewers suggest accepting the paper. While in production, please consider placing emphasis on clarifying the specific contribution of this paper with respect to previous work in other journals. Also please attend to the comment of one reviewer regarding confidence intervals

Reviewer 1 ·

Basic reporting

no comment

Experimental design

no comment

Validity of the findings

no comment

Additional comments

The contents of this paper are very interested and suitable for publishing in this journal.

Reviewer 2 ·

Basic reporting

I am very happy with the revision.

It might be useful to also include the response to my 2. issue into the paper:

2. It should be pointed out how this paper is different from papers in submission with PeerJ already or with Biometrical Journal or other relevant submissions as we do not want to see the same work published twice.
Response: In this submission, confidence intervals for the ratio variances of delta-lognormal are proposed, whereas, other submission proposed CIs for the difference between variances of delta-lognormal populations based on Herbert et al. (2011) and Krishnamoorthy et al. (2011). Although, the difference between them is that the former was considered to estimate the ratio of two delta-lognormal variances (ω_1/ω_2) using HPD-based normal gamma prior. The latter was presented HPDs based on other priors for the difference in delta-lognormal variances (ω_1-ω_2) for large variational situations. According to Herbert et al. (2011) and Krishnamoorthy et al. (2011), the assessment of the range of distribution of (ω_1-ω_2) can be estimated using CIs for (ω_1-ω_2) such that it can be implied that there is the distributed range between (ω_1-ω_2) and (ω_1/ω_2). Finally, both of random variables are used to check significantly difference in the two independent datasets. The next contrast between this submission and other one is the illustrated data that there are different dispersions and locations in each paper.
Remarks:
Herbert, R. D., Hayen, A., Macaskill, P. and Walter, S. D. (2011). Interval estimation for the
difference of two independent variances. Communications in Statistics - Simulation and
Computation 40, 744–758.
Krishnamoorthy, K., Lian, X. and Mondal, S. (2011). Tolerance intervals for the distribution of the
difference between two independent normal random variables. Communications in Statistics –
Theory and Methods 40, 117–129.

Experimental design

no further comments

Validity of the findings

ok

Additional comments

see aboe

Reviewer 3 ·

Basic reporting

I have no other comments.

Experimental design

-

Validity of the findings

-

Additional comments

-